# Prediction of Thermostability of Enzymes Based on the Amino Acid Index (AAindex) Database and Machine Learning

**DOI:** 10.3390/molecules28248097

**Published:** 2023-12-15

**Authors:** Gaolin Li, Lili Jia, Kang Wang, Tingting Sun, Jun Huang

**Affiliations:** 1School of Biological and Chemical Engineering, Zhejiang University of Science and Technology, Hangzhou 310023, China; aug11lgl@gmail.com; 2State Key Laboratory of Rice Biology and Breeding, China National Rice Research Institute, Hangzhou 311400, China; jialili@caas.cn; 3Department of Physics, Zhejiang University of Science and Technology, Hangzhou 310023, China; w826044877@icloud.com

**Keywords:** artificial intelligence, machine learning, thermostability, molecular dynamics simulation, extended sequence, directed evolution

## Abstract

The combination of wet-lab experimental data on multi-site combinatorial mutations and machine learning is an innovative method in protein engineering. In this study, we used an innovative sequence-activity relationship (innov’SAR) methodology based on novel descriptors and digital signal processing (DSP) to construct a predictive model. In this paper, 21 experimental (*R*)-selective amine transaminases from *Aspergillus terreus* (AT-ATA) were used as an input to predict higher thermostability mutants than those predicted using the existing data. We successfully improved the coefficient of determination (R^2^) of the model from 0.66 to 0.92. In addition, root-mean-squared deviation (RMSD), root-mean-squared fluctuation (RMSF), solvent accessible surface area (SASA), hydrogen bonds, and the radius of gyration were estimated based on molecular dynamics simulations, and the differences between the predicted mutants and the wild-type (WT) were analyzed. The successful application of the innov’SAR algorithm in improving the thermostability of AT-ATA may help in directed evolutionary screening and open up new avenues for protein engineering.

## 1. Introduction

In protein engineering, directed evolution is an important method used for modifying the properties of enzymes. It is widely employed in various industries, including chemical or drug synthesis, food production, and waste biodegradation [1,2]. Directed evolution through the natural rule of “mutation selection” remains the most prevalent method for modifying enzymes. Mutant libraries are constructed using procedures such as site saturation mutation, epPCR, and DNA shuffling. They are screened for target mutants that meet specific requirements by recreating the natural evolutionary process in a laboratory [3,4], thus considerably expanding the range of enzyme applications. To date, directed evolution has been used to generate artificial cysteine lipase with high activity and altered catalytic mechanism [5], a redox-mediated Kemp eliminase [6], modified oxidase for efficient CO_2_ fixation [7], and modified transaminase for the synthesis of sitagliptin [8]. However, generating and screening mutant libraries are laborious and time-consuming processes, which dramatically impede the progression of the experiment.

With the development of science and computer technology, machine learning has emerged as an efficient technique for designing novel biocatalysts [9,10,11,12,13]. The use of machine learning methods for predicting the secondary structure of proteins was first reported in 1992 [14]. Since then, many new methods based on machine learning algorithms have emerged for rapid and accurate prediction of the stability, activity, and substrate-binding properties of mutant proteins. For example, support vector machine [15,16] and decision tree [17] can be used to predict changes in the stability of enzymes following mutations, random forest can be used to predict protein solubility [18], and the K-nearest neighbor classifier can be used to predict the function and mechanism of enzymes [19]. However, these methods require the modeling of massive experimental data, and the accuracy of prediction requires to be verified through experiments. Innovative sequence–activity relationship (innov’SAR) is a new method for screening mutant libraries that was reported in 2018 [20]. It predicts the effects of mutations on the biological activity of proteins in a small-sample dataset only using sequence information. The innov’SAR has been successfully used to improve the enantioselectivity of epoxide hydrolase from *Aspergillus niger* [21].

As natural biocatalysts, ω-transaminases play a critical role in the asymmetric synthesis of chiral amines and in the racemic splitting of amines, which are used in the manufacture and synthesis of pharmaceutical products, fine chemicals, and agrochemicals [22,23,24]. The amine transaminase from *Aspergillus terreus* (AT-ATA) has high stereoselectivity and excellent catalytic efficiency [25]; however, its poor thermal stability greatly limits its practical applications. Therefore, improving the thermal stability of transaminases through site-directed mutagenesis is necessary for expanding their industrial applications. We have previously used different protein engineering strategies to improve the thermal stability of transaminases, including the introduction of disulfide bonds, deletion of unstable amino acids on the surface of the loop region, consensus mutagenesis, and the use of B-factor values combined with FoldX energy optimization [26,27]. In addition, we have used machine learning techniques to improve the stability of AT-ATA [28]. With the increasing demand for chiral amines, the further improvement of the thermal stability of AT-ATA can be beneficial for industrial preparation of chiral amines.

In this study, we employed the innov’SAR mutant screening method to build models using various indices, aiming to investigate the impact of protein descriptors on model performance. Furthermore, molecular dynamics simulation analysis was employed to predict distinctions between mutants and the WT.

## 2. Results and Discussion

Figure 1 illustrates the entire process of the innov’SAR method and molecular dynamics simulation; see the Section 3 for details of the methodology. Initially, we gathered experimental data for AT-ATA from wet experiments and encoded them digitally using single-index from the AAindex database [29]. Subsequently, we generated the corresponding protein spectra through FFT. Then, the protein spectral data were utilized to partition the dataset into a learning set and a validation set. The machine learning model was then built by employing the partial least squares regression (PLSR) algorithm. The AAindex database comprised 566 indexes, with 13 of them having missing values. Therefore, we conducted 553 iterations to construct the model, evaluated the root-mean-squared error of cross-validation (cvRMSE) for each index, and incorporated them into the index pool. Next, utilizing information from the index pool, the combined indexing strategy and the iterative indexing strategy were employed for modeling, respectively. This aimed to select the optimal model, which was then combined with the self-constructed library for prediction. Ultimately, mutants with superior thermal stability in the predicted results were selected, and molecular dynamics simulations were conducted to analyze the distinctions between the predicted mutants, experimental mutants, and the WT.

Protein sequences can be encoded by indexing different physical and chemical properties in the AAindex database to generate different digital sequences. In this study, we connected more indices to obtain more biological information about proteins by coding the indices of different physicochemical properties. In the innov’SAR method [30], FFT generates protein spectra that significantly contribute to the performance of the model. Therefore, we only considered the coding sequence after connecting FFT. In the encoding phase, each protein variant of the initial experimental dataset was digitized using the indices in the AAindex database. Subsequently, FFT was used to convert the digital sequence into the corresponding protein spectrum as an input for modeling, and this encoded sequence was denoted as the primary sequence (FFT_Seq). A total of 553 indices are available in the AAindex database for FFT-based encoding of protein sequences. Consequently, 553 FFT_Seqs were generated, which were denoted as FFT_Seq1, FFT_Seq2, FFT_Seq3, and so on. 

Furthermore, the FFT_Seqs were connected, and the connected sequence of numbers was denoted as an extended sequence (Ext_Seq), which served as the new descriptor after modification. Equation (1) represents the construction of the new descriptor, and the “−” symbol indicates the connection of two FFT_Seqs:Ext_Seq = FFT_Seqi − FFT_Seqj − FFT_Seqk…(1)

More information can be obtained using the abovementioned procedure; however, Ext_Seqs can be combined in thousands of ways owing to the redundancy of indices. The formula used to obtain the number of combinations is mentioned below (Equation (2)):(2)$$\sum_{Q=1}^{Q=N}\frac{N!}{Q!(N-Q)!}=2^{N}-1$$

In Equation (2), N represents the number of indices and Q represents the number of FFT_Seqs used to generate Ext_Seq.

Given that the AAindex database has 553 indices, the Ext_Seq set may have 5.4 × 10^165^ possible inputs as modeling codes. Owing to the large number of combinations and considering the performance of the computer, the size of the Ext_Seq set was reduced by filtering a smaller number of FFT_Seqs for combination and identifying the best Ext_Seq by evaluating the modeling performance. We analyzed the following two strategies to construct Ext_Seq: (1) using a single-index encoding the top five elementary indices for combination and (2) identifying the best Ext_Seq through an iterative connection strategy.

### 2.1. Combination Strategy Using Multiple Index Codes

To limit the size of the Ext_Seq set, we restricted *N* to the top five best indices and defined the extended sequence to have a maximum of five elementary sequence connections, that is, *Q* ≤ 5. The construction of the extended sequence was independent of the order of elementary sequence connections, that is, FFT_Seq1 − FFT_Seq2 is equivalent to FFT_Seq2 − FFT_Seq1. Initially, we utilized Dataset 1 with a single-index coding approach. By evaluating the cvRMSE of the model, we selected the top five indices and used them for combination. Table 1 shows detailed information on the top five indices, including the title, serial number, cvRMSE, and R^2^ of the indices. According to Equation (2), there are 31 combinations of Ext_Seq. We used each Ext_Seq as a modeling input and ranked the sequences according to R^2^, the parameter indicating the predictive performance of the model, to identify the best combination of Ext_Seq.

Table 2 shows the top 10 extended sequences based on the top five index combinations. Compared with the R^2^ and cvRMSE of the previous single-index coding model, the R^2^ of the best extended-index model was increased from 0.81 to 0.86 and its cvRMSE was decreased from 4.56 to 3.64 (Figure 2a,b). The R^2^ of the previous best single-index AURR980108 model ranked 12th in terms of the combined indices. From the perspective of model performance, the combination of the top-ranked indices resulted in a certain improvement in the performance of the model for the extended sequence. 

Furthermore, when the top five indices were concatenated, a longer extended sequence was generated: Ext_Seq = FFT_Seq1 − FFT_Seq2 − … FFT_Seq5. The cvRMSE and R^2^ of this extended sequence were 4.76 and 0.80, respectively (Figure 2c), ranking 13th in terms of overall ranking, with a reduced performance compared with the best model with single-index coding. Therefore, the length of the extended sequence was not as long as possible. Moreover, the best single-index AURR980108 model was ranked 12th in terms of overall ranking, and the combination of m indices was not better than the combination of n indices (m > n). Based on the abovementioned problems, the combination strategy for selecting the top-ranked indices was not the optimal way to obtain the best Ext_Seq, although the combination strategy improved the predictive performance of the model. Therefore, we used an iterative connection strategy to identify the best Ext_Seq and the best combination of Ext_Seq cases [30].

### 2.2. Iterative Connection Strategy

The iterative connection strategy [30] is used to identify the best index for the next connection by continuously expanding the size of Ext_Seq. For each iteration, the current best sequence index (Ext_Seq) was retained through the leave-one-out cross-validation (LOOCV), which enabled the construction of Ext_Seq using different index increments. Simultaneously, the best index of modeling performance at the end of the current iteration was determined and prepared for the next iteration.

Figure 3 demonstrates the whole iteration process. After single-index modeling is complete, we select the best FFT_Seq through the index pool. With the initial construction of 553 models, we use cross-validation to assess and rank the cvRMSE of each model, and finally select the index with the lowest cvRMSE in the index pool as the best index. Following that, the best index is utilized to connect with each of the remaining 552 indexes in the index pool, resulting in the generation of 552 extended sequences. The best extended sequence is retained for the next connection based on cvRMSE evaluation, and the number of indexes in the index pool is reduced by one after each connection. The iteration continues sequentially, consistently searching for a suitable sequence of extensions. 

Given that the iterative connection strategy can inevitably increase the time complexity of the algorithm, we limited the number of connection indices to three (*Q* ≤ 3): Ext_Seq = FFT_Seq1 − FFT_Seq2 − FFT_Seq3. Figure 2a demonstrates the best model obtained through single-index modeling based on the experimental half-life data of 13 AT-ATA with R^2^ and cvRMSE of 0.81 and 4.56, respectively. Figure 4 shows the results of applying the iterative connection strategy to the abovementioned dataset, with R^2^ and cvRMSE of 0.96 and 1.93, respectively. Based on the parameters used to evaluate the performance of the model, we concluded that the iterative connection strategy greatly improved the quality and predictive performance of the model with only three indexed connections. Therefore, we increased the sample size by including the new AT-ATA experimental Dataset 2 to assess whether the model could predict variants with a higher half-life than the current experimental best half-life. 

### 2.3. Prediction of Newly Improved AT-ATA Mutants

The previously described best experimental mutant F115L_L118T harbors two single-point mutations in its sequence with a half-life of 65.9 min [26]. To identify better mutants, we constructed a new prediction model using Dataset 2, combined with the iterative connection strategy, which was named model_21. Compared with the R^2^ of the single-index strategy, the R^2^ of the new model was successfully improved from 0.66 to 0.92 and all combinations of the 10 single-point mutations (2^10^ variants) were computationally generated. The method generated 1003 (2^10^ − 21 = 1003) new variants with multiple-point mutations. Figure 5 shows the predicted half-life of all variants in model_21, with 265 mutants having a higher half-life than F115L_L118T, indicating that that model can identify candidates with better thermal stability than F115L_L118T.

### 2.4. Analysis of MD Simulation for Predicting AT- ATA Mutants, Experimental Mutants, and WT

Among the above predictions, there are two mutants with half-lives larger than 100 min. We named them P1 and P2. Details of the mutation sites and predicted half-lives of the two mutants P1 and P2 are provided in Table 3. We simulated the P1, P2, F115L_L118T, and WT structures for up to 100 ns by molecular dynamics using YASARA.

RMSD is a metric that gauges the average deviation between the protein conformation and the initial structure at a given time. It serves as a crucial indicator for assessing whether the system has reached equilibrium. As depicted in Figure 6, the systems of P1, P2, and F115L_L118T have attained equilibrium at the 100 ns simulation stage. The system of WT deviates significantly from the initial conformation of the protein. However, P1 and P2 show smaller deviations from the initial conformation. In addition, based on the last 20 ns of the trajectory, the RMSDs of WT, F115L_L118T, P1, and P2 were stable at 0.27, 0.23, 0.2, and 0.18 nm, respectively. It is noticeable that P2 exhibits less deviation from the initial conformation then P1. However, the prediction indicates P1 has more superior thermal stability than P2. This suggests that the correlation between RMSD and thermal stability might not be very strong. Nevertheless, for both mutants P1 and P2, the systems reached equilibrium more rapidly than WT.

RMSF is the root-mean-squared displacement of each amino acid in a given conformational frame compared to the average conformation, revealing the dynamic nature of the protein structure. Due to distinct biological functions in different regions of a protein, observing RMSF changes in residues allows for the identification of flexible regions and key residues in the structure. 

Figure 7 demonstrates the fluctuation levels in the residues of the four systems. AT-ATA is a dimeric structure containing two chains, A and B. As can be observed from Figure 7a, the residue fluctuation level of WT-A is overall higher than that of F115L_L118T-A, P1-A, and P2-A, and the average RMSF values decrease sequentially. This suggests a relatively more flexible protein structure in WT-A. On the contrary, in Figure 7b, the difference in the level of residue fluctuations between WT-B, F115L_L118T-B, P1-B, and P2-B is small. Despite being in the same system, the two chains exhibit different fluctuations, possibly attributed to the incomplete symmetry of AT-ATA. Additionally, by observing the peaks of the two plots, it can be noticed that in each system, residues Thr23, Arg131, and Arg232 present greater fluctuations with higher RMSF values relative to the surrounding amino acid residues.

SASA is commonly used to reflect the solvent-exposed surface area of proteins, and a variation of this parameter can reveal the dynamic nature of the protein structure. In general, the smaller SASA indicates that the protein molecule is in a more folded and compact structural state. In Figure 8, we clearly observe that the SASAs of P1 and P2 are overall smaller than that of WT and F115L_L118T, indicating that the structures of P1 and P2 are more compact. As the simulation proceeds, considering the last 20 ns of the trajectory, the average SASA values for WT, F115L_L118T, P1, and P2 are 293.8, 289.5, 288.1, and 287.5 nm^2^, respectively. It suggests that the structures of P1 and P2 are more compact than WT and F115L_L118T. 

Figure 9 illustrates the fluctuation in the number of hydrogen bonds between the A and B chains of the molecules in the four systems throughout the simulation. Hydrogen bonding can provide insights into the structure and interactions of proteins. Typically, a higher number of hydrogen bonds may imply a more stable protein structure. During the 100 ns simulation, the number of hydrogen bonds in the four systems kept changing, indicating that the simulation process was causing the breaking of hydrogen bonds and the formation of new ones.

The average number of hydrogen bonds for WT, F115L_L118T, P1, and P2 were 521, 526, 538, and 543, respectively, showing a sequential increase. The number of hydrogen bonds for P1 and P2 was higher than that of WT and F115L_L118T, indicating that the protein structures of mutants P1 and P2 were more stable than that of WT and F115L_L118T. 

Figure 10 illustrates the variation of the radius of gyration for the four systems. The radius of gyration is a physical quantity that describes the structural compactness of a protein and provides information about the state of protein folding. A smaller radius of gyration may mean that the protein is more compact and folded, while a larger radius of gyration may indicate that the protein is more unfolded or loose. In Figure 10, the radius of gyration of WT is overall higher than that of F115L_L118T, P1, and P2, indicating that the WT structure is more loosely packed relative to F115L_L118T, P1, and P2. In addition, according to the last 20 ns of the trajectory, the radius of gyration for WT, F115L_L118T, P1, and P2 stabilized at 27.07, 26.95, 26.9, and 26.86 Å, respectively. This follows a sequentially decreasing order, similar to the patterns observed in RMSD and SASA.

It is not difficult to observe that the stability of P2 is slightly better than that of P1, although this is contradicted by the fact that P1 has been predicted to have a stronger thermal stability than P2. However, the thermal stability distinction of P1 and P2 is not obvious since the predicted half-life values of P1 and P2 have not much difference. Nonetheless, it is found that the stability of P1 and P2 was much higher than that of WT and F115L_L118T, suggesting that the innov’SAR remains an effective method in AT-ATA screening for mutants with improved thermal stability.

## 3. Materials and Methods

### 3.1. Aspergillus terreus Dataset

The experimental dataset was acquired from the previous work by Huang et al., and it was divided into two groups. As shown in Table 4, Dataset 1 was used to test whether the innov’SAR model was improved, and Dataset 2 was used to predict more thermally stable mutants from the latest experimental Dataset 2. Dataset 1 included a collection of sequences of wild-type (WT) AT-ATA and its 12 mutants. Their half-lives (*T*_1/2_) are also included. *T*_1/2_ was used to measure the thermal stability of enzymes and was defined as the time required for the residual activity of AT-ATA to decline to 50% of its initial activity at 40 °C.

### 3.2. innov’SAR

Figure 11 demonstrates the workflow of the innov’SAR method [30]. The whole process was divided into three phases: encoding, modeling, and prediction phases. During the encoding phase, the protein variants were digitally encoded using different indexes in the AAindex database to generate different digital sequences, which were transformed into the corresponding protein spectra in combination with FFT. In digital signal processing, the most critical step, FFT converts the digital signals to a representation in energy and frequency domains [31,32] (Equation (3)). Based on the different protein spectra, Ext_Seq was generated via concatenation to obtain more biological information about the protein sequence. Subsequently, in the modeling stage, Ext_Seq and half-life are used to find the best prediction model in combination with PLSR. Finally, in the prediction stage, the mutant libraries are put into the prediction model and the half-life of each protein variant is predicted.
(3)$$S(k)=\sum_{n=0}^{N-1}s(n)e^{\left(-2i\pi \frac{n}{N}\right)}$$

In Equation (3), s is the input signal of length N (amino acid sequence), S is the input spectrum (imaginary unit), k is the frequency in the spectrum, n is the position in the input signal, and i is the complex number with i^2^ value of −1.

### 3.3. Evaluation of Modeling Performance

During the modeling phase, the model was evaluated based on the root-mean-squared error of cross-validation (cvRMSE) and the coefficient of determination (R^2^). The leave-one-out cross-validation (LOOCV) approach was used in this study. cvRMSE was estimated to select the optimal model because it indicated the degree of variation in prediction while using different training sets. R^2^ was estimated to evaluate the predictive ability of the model, reflecting the degree of concordance between the experimental and predicted half-life. The formulas used for calculating cvRMSE and R^2^ are mentioned below (Equations (4) and (5)):(4)$$cvRMSE=\sqrt{\sum_{i=1}^{S}\frac{{\left(y-{\hat{y}}_{}\right)}^{2}}{S}}$$
(5)$$R^{2}=\frac{{\left(\sum_{i=1}^{S}\left(y_{i}-\bar{y})({\hat{y}}_{i}-\hat{\bar{y}}\right)\right)}^{2}}{\sum_{i=1}^{S}{\left(y_{i}-\bar{y}\right)}^{2}\sum_{i=1}^{S}{\left({\hat{y}}_{i}-\hat{\bar{y}}\right)}^{2}}$$

In the abovementioned equations, $y_{i}$ is the experimental half-life of the *i*th sequence, ${\hat{y}}_{i}$ is the half-life of the *i*th sequence predicted using the ISAR method, $\bar{y}$ is the average of the experimental half-life, and S is the number of sequences.

### 3.4. Molecular Dynamics Simulation

To verify the validity of the predictive model, we obtained the crystal structure of AT-ATA (PDB ID: 4CE5) from the Protein Data Bank “http://www.rcsb.org (accessed on 11 August 2022)” and used molecular dynamics simulation to compare the thermal stability of mutants and WT. Because only the effect of mutation on protein stability was considered, the water of crystallization, impurity ions, and related substrates of AT-ATA were removed during the pretreatment stage. Based on the initial crystal structure, we additionally generated three AT-ATA mutant proteins: the optimal mutants P1 and P2 through model screening, and the experimentally measured optimal mutant F115L_L118T. The three-dimensional (3D) structure of AT-ATA mutants was homology-modeled and optimized using the “BuildModel” command of the FoldX (version 5.0) software [33,34].

The simulation process was implemented using the YASARA (version 16.4.6) software [35] “http://www.yasara.org (accessed on 11 August 2022)”, with the Amber14 force field set for all systems and a constant temperature of 313 K for 100 ns MD simulations. First, the protein was placed in a cube with a density of 0.998 mg/L, and the whole cube was filled with water with a density of 0.998 mg/L. Sodium and chloride ions (0.9%) were added to neutralize the system charge to ensure correct osmotic pressure and electrostatic neutrality, and the ionizable group was protonated according to the PKA value of pH 8.0 in the medium. Subsequently, the gradient descent method was used to minimize the energy of the system. Finally, 100 ns MD simulations with a step size of 2.5 fs were completed under the condition of constant temperature and pressure, and the trajectory file was saved every 25 ps.

During simulation, the cutoff value for van der Waals forces and electrostatic interactions was set to 8.0 Å. At the end of the simulation, YASARA was used to analyze the overall and local changes in the structure of each mutant protein during the simulation. These changes included root-mean-squared deviation (RMSD) of backbone atomic positions, root-mean-squared fluctuation (RMSF) of individual residues, solvent accessible surface area (SASA), hydrogen bond number, and gyration radius.

## 4. Conclusions

In this study, we successfully used the innov’SAR method for the rational screening and modification of thermostability AT-ATA. As an efficient combinatorial mutant library screening tool, the half-lives of 1024 mutants were predicted using an experimental dataset that accounted for only 2% of the total mutation combinations, reducing the burden of experimental screening. In addition, the innov’SAR method was used in combination with the iterative connection strategy to obtain Ext_Seq, and the R^2^ of the model was improved from 0.66 to 0.92, indicating substantial improvement in the prediction performance of the model. Finally, we applied molecular dynamics simulation to analyze the distinctions between mutants and the WT. Mutants P1 and P2 are verified that they have more thermal stability than WT and F115L_L118T. It offers insightful observations, paving the way for further exploration in mutation studies.

The greatest advantage of the innov’SAR method is that it only requires protein sequence information combined with experimental characteristic values and not the spatial structure information of proteins. The method can rapidly predict the characteristic values of combinatorial mutant libraries based on single-point mutations and can be used to screen for beneficial mutants. The protein sequences containing non-standard amino acid compositions remains elusive. Moreover, at present, the innov’SAR screening tool is only an in silico scripting program. Therefore, the numerical handling of non-standard amino acids should be further elucidated and a publicly accessible web server should be developed so that the innov’SAR method can be applied to the directed evolution of various enzymes, thereby opening up new avenues for protein engineering.

## Figures and Tables

**Figure 1 molecules-28-08097-f001:**
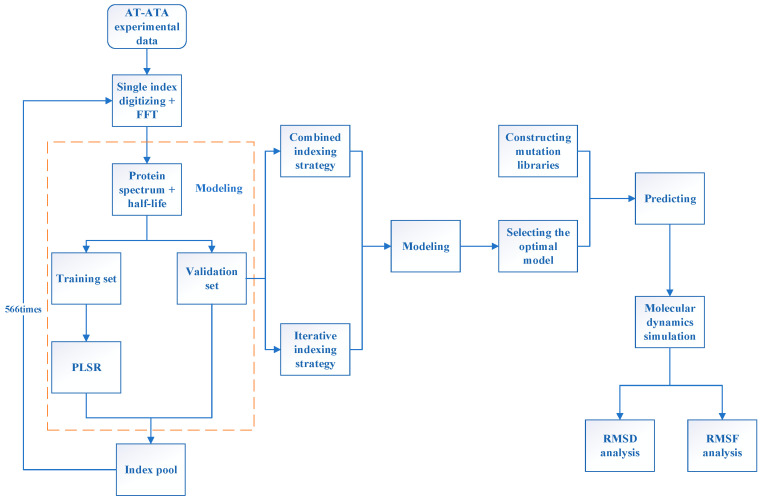
The overall flowchart of the innov’SAR method and molecular dynamics simulation.

**Figure 2 molecules-28-08097-f002:**
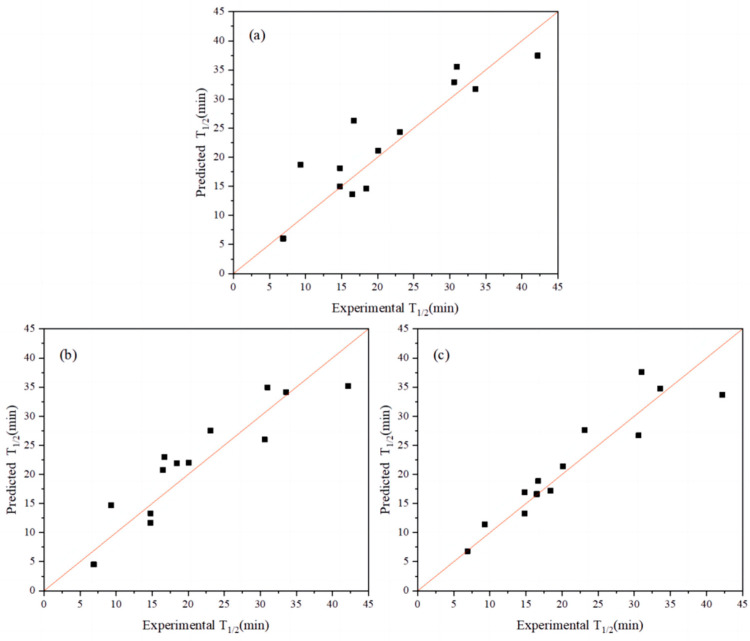
Thermal stability plots of measured and predicted half-lives of AT-ATA variants. (**a**) Use of a single index: R^2^ = 0.81. (**b**) The optimal combination of indices: R^2^ = 0.86. (**c**) Connection of the top five single-index combinations in series: R^2^ = 0.80.

**Figure 3 molecules-28-08097-f003:**
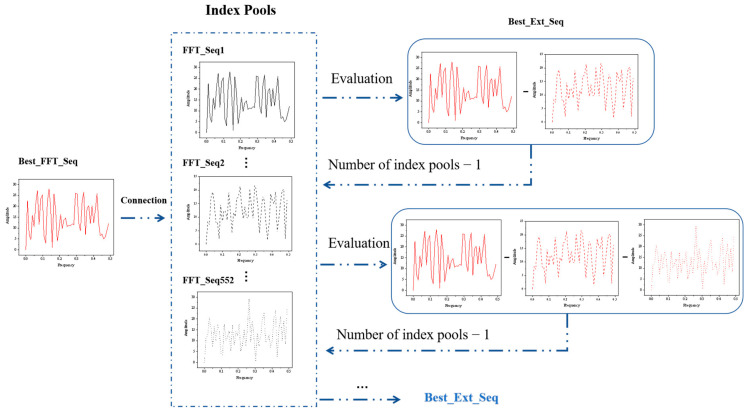
Flow chart of the iterative process of successive concatenation. Each round uses the indices of the previous iteration as the basis for the extended sequence and determines the best index to retain for the current round by evaluating the performance of the model.

**Figure 4 molecules-28-08097-f004:**
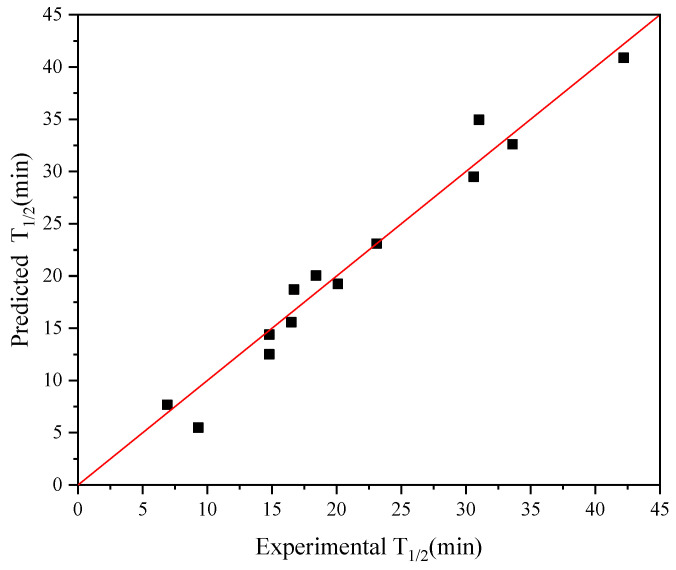
Plot for experimental versus predicted thermal stability of AT-ATA variants. The graph was plotted using iterative connect indices (AURR980108-MEIH800103-CORJ870104): R^2^ = 0.96.

**Figure 5 molecules-28-08097-f005:**
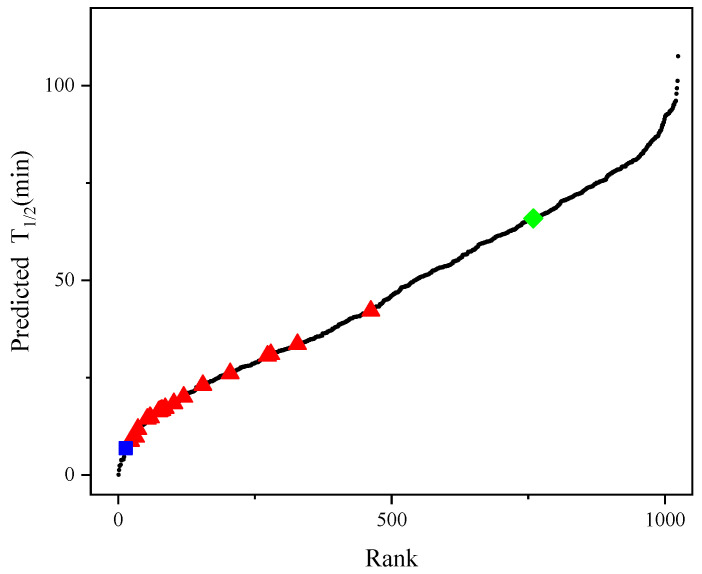
Half-lives of the 1024 possible variants of AT-ATA. (■): half-life measured for WT, (◆): half-life measured for the best experimental mutant F115L_L118T, (▲): half-life measured for the remaining single and multi-site mutants, (●): predicting half-life of all 1024 possible variants.

**Figure 6 molecules-28-08097-f006:**
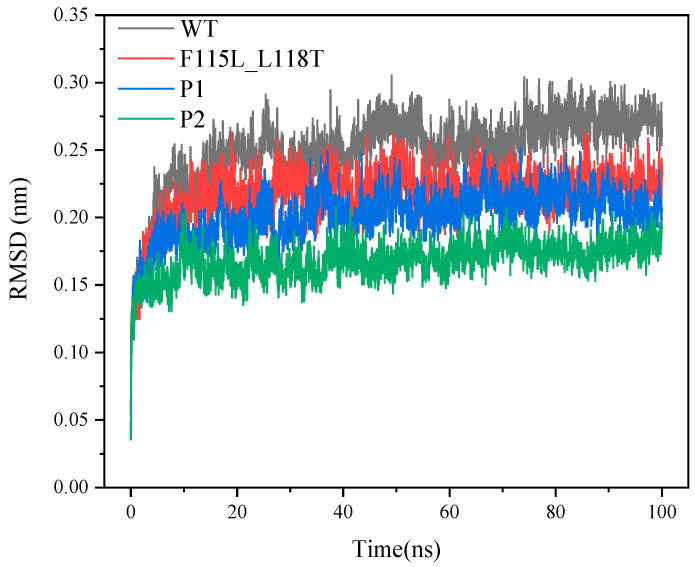
RMSD values of P1, P2, F115L_L118T, and WT in 100 ns simulations.

**Figure 7 molecules-28-08097-f007:**
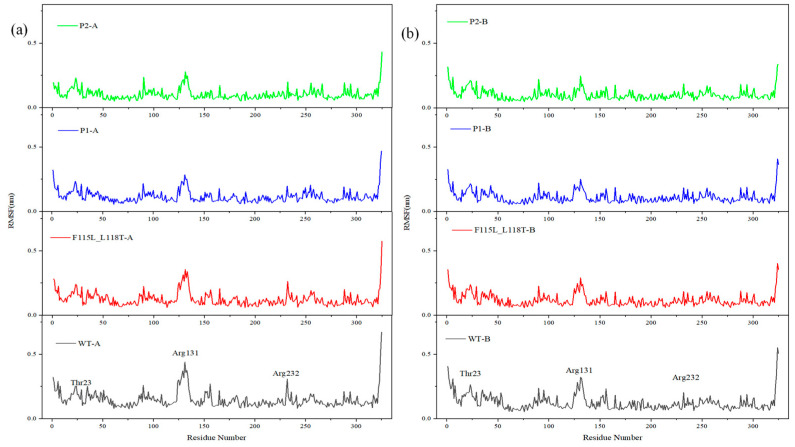
MD analysis of P1, P2, F115L_L118T, and WT using YASARA at 313 K in the last 20 ns. (**a**) RMSF of P1-A, P2-A, F115L_L118T-A, and WT-A. (**b**) RMSF of P1-B, P2-B, F115L_L118T-B, and WT-B.

**Figure 8 molecules-28-08097-f008:**
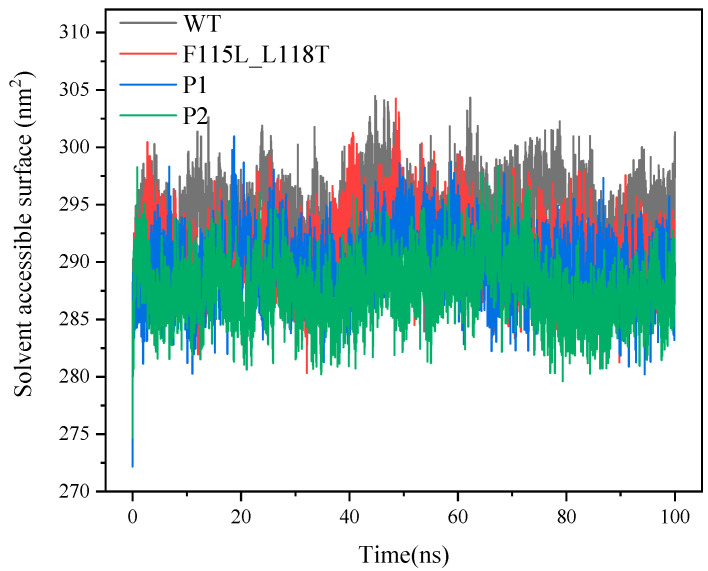
SASA values of P1, P2, F115L_L118T, and WT in 100 ns simulations.

**Figure 9 molecules-28-08097-f009:**
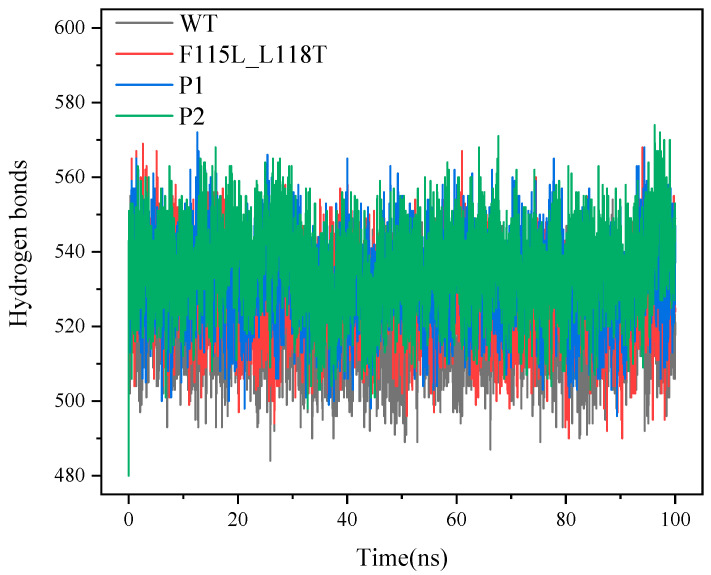
The number of hydrogen bonds between the A and B chains of the P1, P2, F115L_L118T, and WT in 100 ns simulations.

**Figure 10 molecules-28-08097-f010:**
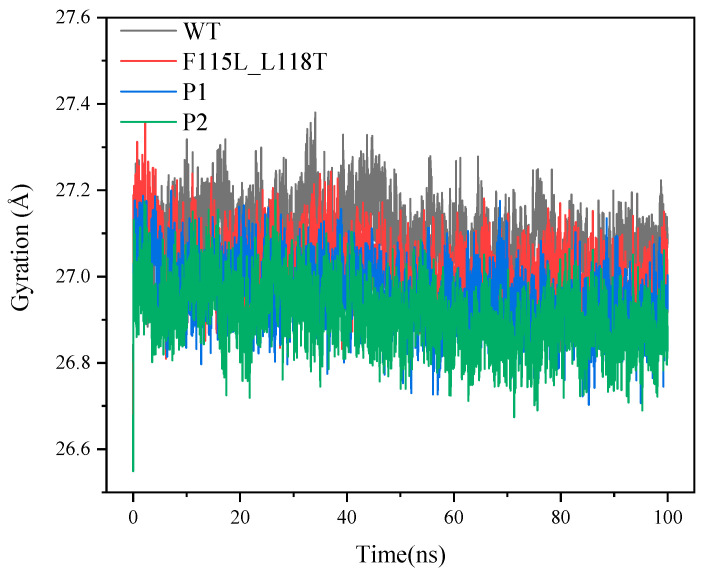
The radius of gyration values of P1, P2, F115L_L118T, and WT in 100 ns simulations.

**Figure 11 molecules-28-08097-f011:**
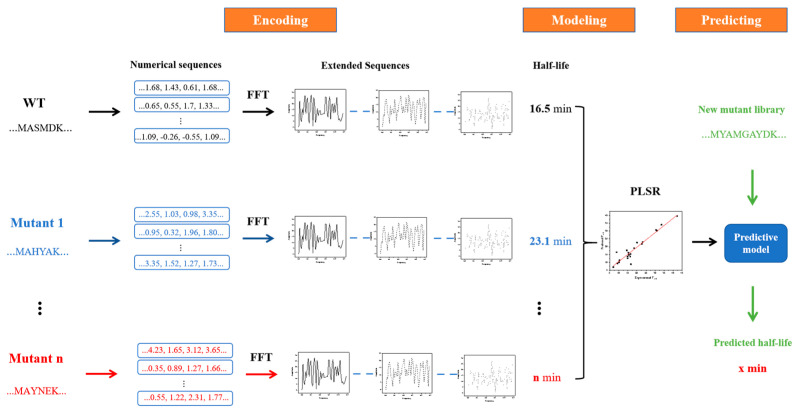
Schematic diagram of the innov’SAR method with extended sequences.

**Table 1 molecules-28-08097-t001:** The top five indices encoded by a single index.

Index	Index Number	cvRMSE	R^2^
AURR980108	396	4.56	0.81
OOBM770102	201	5.33	0.76
MUNV940102	416	6.01	0.67
CORJ870102	507	6.10	0.67
GEOR030108	484	6.58	0.59

**Table 2 molecules-28-08097-t002:** The top 10 extended sequences selected based on the combination of the top 5 indices.

Index Number	cvRMSE	R^2^
396,201,507,484	3.64	0.86
396,507,484	3.71	0.86
396,201,507	3.91	0.85
201,507	4.36	0.84
396,416,507	4.22	0.84
396,201,416,507	4.17	0.83
201,507,484	4.20	0.83
396,201	4.25	0.82
396,507	4.33	0.82
396,201,416	4.37	0.81

**Table 3 molecules-28-08097-t003:** The mutation site and predicted half-lives of the optimal screened mutants.

Variant	Mutations	Predicted *T*_1/2_ (min)
P1	Q97E_F115L_L118T_E133A_H210N_N245D_E253A_G292D	107.59
P2	I77L_F115L_L118T_E133A_H210N_N245D_E253A	101.25

**Table 4 molecules-28-08097-t004:** AT-ATA experimental dataset.

Mutations	*T*_1/2_ (min)	Note
WT	6.9	Dataset 1/Dataset 2
I77L	20.1	Dataset 1/Dataset 2
Q97E	16.5	Dataset 1/Dataset 2
F115L	17.2	Dataset 2
L118T	26.1	Dataset 2
E133A	9.8	Dataset 2
H210N	23.1	Dataset 1/Dataset 2
N245D	14.8	Dataset 1/Dataset 2
E253A	11.8	Dataset 2
G292D	14.8	Dataset 1/Dataset 2
I295V	9.3	Dataset 1/Dataset 2
F115L_L118T	65.9	Dataset 2
I77L_H210N	42.2	Dataset 1/Dataset 2
Q97E_H210N	30.6	Dataset 1/Dataset 2
H210N_N245D	18.4	Dataset 1/Dataset 2
H210N_G292D	33.6	Dataset 1/Dataset 2
I77L_Q97E_H210N	31	Dataset 1/Dataset 2
I77L_H210N_G292D	16.7	Dataset 1/Dataset 2
I77L_Q97E_H210N_N245D	14.4	Dataset 2
I77L_H210N_N245D_G292D	16.3	Dataset 2
I77L_Q97E_H210N_N245D_G292D	8.7	Dataset 2

## Data Availability

Data are contained within the article.

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
