# Peer review of "Prediction of Thermostability of Enzymes Based on the Amino Acid Index (AAindex) Database and Machine Learning"

_molecules, 2023, doi:10.3390/molecules28248097_

Round 1

Reviewer 1 Report

Comments and Suggestions for Authors

Here, I review the manuscript entitled „Prediction of Thermostability of Enzymes Based on The Aaindex Database and Machine Learning“, which was submitted to Molecules. In their own words, the authors present an improved innov‘SAR algorithm, with which they predict the thermostability of mutations of (R)-selective amine transaminases from Aspergillus terreus (AT-ATA). While the paper is generally well-readable, I recommend rejecting the manuscript in the current form. I would recommend the authors major rewriting and resubmission to a more specialized journal. Below, I explain my reasons for this recommendation.

My biggest point of criticism is that the authors, over a wide range of the manuscript, stay *very* close to the original publication of innov‘SAR. Generally, I want to acknowledge that the authors cite the original publication of innov‘SAR. However, the similarities between these two manuscripts are really quite remarkable: Many figures are almost identical, for example figures 1 of both manuscripts, but also others. Furthermore, essentially all equations are the same. Moreover, some paragraphs may be read in both manuscripts, sentence by sentence, with essentially the same content.

Certainly, this creates a large redundancy. As a result, it is difficult to identify the differences between the original approach and their work. I would strongly recommend that the authors rewrite their manuscript and, in the process, try to minimize redundancies and thereby emphasize their own approach and findings. 

Besides, I would also suggest to illustrate their improvement of the original method, by comparing with the dataset of the original publication. Showing an improvement of the method on the same data as originally published would be even more convincing. (Just a suggestion.)

I acknowledge that their approach to use MD simulations as a complementary method is indeed an additional effort in comparison to the original publication of innov‘SAR. I have a few points considering their implementation of this approach:

1. Showing differences in the RMSD does not generally explain thermostability. An example: Looking at Fig 6. I expect, P1 to be in slightly different conformation than P2 at the end of the simulation. The authors may show an overlay of the final structures to visualize these slightly different conformations. However, if (for example) a loop between two helices is in different conformation in such a short simulation, this does not allow any conclusions about thermostability, because these slightly different conformations may still be equally stable. Actually, the conformation with a higher RMSD to the crystal structure may even be more stable…

2. It would be interesting to include a comparison to the wildtype, e.g., considering the RMSD, but also RMSF.

3. The authors decided to show the average RMSF value for every five residues. I would suggest showing the RMSF of every single residue. (Potentially, only in the SI). Otherwise, it might be possible that they wash out some local fluctuations.

3. The authors hypothesize that the Mutation E133A leads to improved thermostability. One of the authors has already formulated this hypothesis in an earlier publication. They cite their prior work, however, they could have indicated even clearer that they already looked at this particular point mutation before and came to exactly this conclusion. I noticed that the experimental results in Table 4 actually show hardly any improvement of the thermostability of E133A in comparison to other mutations. This seems to contradict their hypothesis. I would suggest the authors discuss this matter in more detail.
Furthermore, the authors claim that a double hydrogen bond was formed between ARG131, ALA133 and ASP134. However, in Figure 7, I cannot see this arrangement of hydrogen bonds. Eventually, the authors may elaborate more on this hydrogen bond network. Do they mean the backbone of Ala? (The side chain of Ala should not be involved in hydrogen bonds...)

Reviewer 2 Report

Comments and Suggestions for Authors

I have only minor comments to improve the manuscript. See attached.

Reviewer 3 Report

Comments and Suggestions for Authors

Overall, the article was well-written and I believe the machine learning part of the experimentation was done well. I also believe the construction of the simulations was done well.

However, I do not believe the analyses of the MD simulations or the number of simulations conducted are sufficient to function as validation of the ML prediction in lieu of any other experimental evidence.

The authors evaluated a single known variant and two predicted variants with MD. At minimum, they should additionally test WT, 1 variant with experimentally known lower thermal stability and 1 variant with in silico predicted lower affinity. What should be seen is a rank ordering that is in agreement with experimental data, i.e. less stable variant < WT < more stable variant, and the predicted variants should also fit in a rank ordering.

There are programs and plugins to look at predicted stability directly for each frame of the simulation or from discrete frames taken at regular intervals (e.g. every ns). This is the single most relevant in silico analysis of the simulation data for the study, yet it wasn't done.

RMSD and RMSF alone are not direct indicators of thermal stability. They show motion of the protein or residues, which is related to entropy. All that the data imply is the P2 variant adopts a conformation that is closer to the starting structure and that the P1 and F115L-L118T variant adopt a conformation that are probably similar to each other. Notably, the P1 and F115L-L118T have nearly identical RMSD plots and have half-lives the furthest apart (107.59 and 65.9); they are also more similar on the RMSF plot. These data are telling me that there is no relationship between predicted affinity, real affinity, and RMSD/RMSF, the opposite of what the authors conjecture.

As the authors mentioned, MD simulations sample conformations. How was a reference frame chosen to investigate what appears to be a single frame inspection of only a handful of H-bonds? There are easy tools (e.g. VMD) to look at the total number of H-bonds per frame, as well as the % of frames which each H-bond occurs. These would improve the utilization of the simulation data and make them more relevant to the study. As it is, the simulations seem like afterthought.

Other analyses like radius of gyration, SASA, secondary structure changes over time, etc are all easily implemented and can help bring corroboratory data that give a consensus of thermal stability.

Additionally, a large portion of stability comes from other interactions. Salt bridges, pi-pi, hydrophobic, etc. These can all be investigated through programs like VMD fairly easily. It would add more depth to the study and make the simulation data more relevant.

Alternatively, the authors could simply validate some of the predicted variants with physical experimentation.

I fully support the usage of MDS as validation of other in silico data, but it must be constructed and performed properly (which appears to be done) and also analyzed thoroughly to bring sufficient evidence to serve as validation (which hasn't been done here).

Round 2

Reviewer 3 Report

Comments and Suggestions for Authors

I agree with the authors' assessment that the RMSD (fig 6) does not correlate well with thermal stability. While with P1 and P2 it is only predicted thermal stability (not measured), the variant which was removed in the new draft (F115L-L118T) and WT both had experimental half-lives and the RMSD data do not corroborate a ~10-fold difference in half-life. Since the F115L-L118T data were excluded from the current draft, none of the newer simulation metrics are available for our comparison. These data should be brought back and the newer parameters evaluated and compared between WT, F115L-L118T, P1, P2, and it would be best to include other variants with experimental T1/2 as well.

I would also make the same assertion of disagreement for RMSF (fig 7) where the differences between A and B chain for the same simulation seem to show bigger differences than the variants show from each other. There appears to be little to no pattern that suggests RMSF is related to predicted thermal stability.

The newer measurements (SASA, H-bonds, and RoG), based on my own suggestions, were more compelling in the sense that P1 and P2 were similar to each other, and that they were showing the trend expected compared to WT. An interesting thing is that RoG for WT seems to decrease around 40ns, at odds with the expectation if it were beginning to unfold or be unstable.

That being said, the half-life for WT is ~7 minutes and P1 and P2 were predicted to be >100 minutes. The simulation data are overall showing quite minor differences between the variants and the WT, which is not suggestive of drastic differences in stability. Given that the MD data were supposed to serve as validation of the in silico predictor, I do not think this goal is accomplished.

Why were the F115L-L118T data removed since last version? Given this was a variant that had an experimental T1/2, along with WT, this is the only link to reality. If this variant compared to WT shows the expected trends, that is the strongest piece of evidence the authors can supply currently that shows there is a relationship between actual half-life, predicted half-life, and behavior in the simulation. Then and only then can the simulations be extended to predicted variants as partial validation of the in silico predictor.

We need to be cautious when in silico data are being validated with other in silico data. This being done poorly is the exact reason why in silico data has a poor reputation and is not widely trusted. 

The premise of the paper is they employed a pre-existing algorithm innov'SAR and show they can use it to generate variants with improved thermostability. They conclude that the algorithm is effective at identifying more stable AT-ATA variants. However, they make no direct measurement of stability and the simulation data are not showing large differences that would be expected for drastic improvements in stability. Either the innov'SAR algorithm is not accurate here, the simulations are too short to effectively test the stability (100ns simulations when the half-life is in the order of minutes), or the measurements they are making are not able to distinguish stability with good resolution.

The authors must directly compare simulations of AT-ATA WT and variants which have experimentally measured T1/2 and see behaviors that corroborate expectation and magnitude compared to real stability. Once they demonstrate simulations on their system can agree with reality, only then can we trust that it is useful in determining stability in this context and can then be implemented as a validation system.
